# Gender disparities and barriers to access and use of essential health services in Ethiopia: Designing primary health care through gender lens

**Agumasie Semahegn**[1]*, **Yohannes Mehretie Adinew**[2], **Gizachew Tadele Tiruneh**[3], **Mesele Damte Argaw**[1], **Biruhtesfa Bekele**[3], **Metadel Tesfaye**[3], **Nebreed Fesseha**[3], **Samrawit Hagos Tiku**[4], **Asmeret Gebrehiwet**[1], **Addis Girma**[1], **Mebrie Belete**[1], **Chala Tesfaye**[3], **Mikiyas Teferi**[1], **Hillina Tadesse**[3], **Nuru Hussien**[4], **Anne Austin**[3], **Salsawit Shiferraw**[1], **Sintayehu Abebe Woldie**[1,5], **Muluken Dessalegn Muluneh**[1], **Lidiya Tefera**[6], **Frank DelPizzo**[6], **Addis Tamire**[1], **Temesgen Ayehu**[1], **Dessalew Emaway**[2], **Misrak Makonnen**[1]

**1** Amref Health Africa in Ethiopia, Addis Ababa, Ethiopia, **2** EngenderHealth Ethiopia, Addis Ababa, Ethiopia, **3** JSI, Addis Ababa, Ethiopia, **4** DAB Development Research and Training P.L.C, Addis Ababa, Ethiopia, **5** Nossal Institute for Global Health, Melbourne School of Population and Global Health, The University of Melbourne, Melbourne, Australia, **6** Gates Foundation, Addis Ababa, Ethiopia,

* agumas04@gmail.com

## Abstract

Despite remarkable gains in maternal and child health (MCH) outcomes globally, the lack of access to and use of essential health services remains a challenge in low-income countries, like Ethiopia. To address health service access and use, the Ethiopian government has endorsed a fifteen-year "Optimizing Ethiopian Health Extension Program" strategy. This study explored gender barriers to access to and use of MCH services in the primary health care (PHC) setting in Ethiopia. An exploratory qualitative study was conducted in nine districts in Ethiopia. Data were collected using 87 key informants and 134 discussants in 18 Focus Group Discussions. Maximum variation sampling use used to identify participants.. Collected data were coded and gender-related barriers were categorized using the USAID's gender analysis domains. The study enrolled 221 study participants who represented key stakeholders, and 45.7% of them were female. Although Ethiopia adopted and ratified global human rights conventions to protect the legal rights and status of women and children, there are gaps in institutional practices of laws, policies, and regulations. Socio-cultural norms favor men and boys over women and adolescent girls which adversely affect their access to and control over assets and resources. Power and decision-making imbalances between men and women are exacerbated by low men's understanding of MCH needs. Women are not empowered to utilize exempted services. Women's traditional workload in the household during and after pregnancy is prioritized over healthcare. Women face challenges in accessing and use of health

**Data availability statement:** All related qualitative data are presented fully within the paper.

**Funding:** This study was supported by the Bill & Melinda Gates Foundation (BMGF) (INV-002643 to MM and INV-037995 to DE). The funders had no role in study design, data collection and analysis, decision to publish, or preparation of the manuscript.

**Competing interests:** The authors have declared that no competing interests exist.

services due to socio-cultural norms, beliefs, and implementation gaps within the health system of Ethiopia. It is suggested to work with agrarian and pastoral communities to overcome socio-cultural, knowledge-based gender barriers, and addressing gender gapsthat limit women's access to available health services while institutionalizing and tailoring national gender-integrated policies to improve access to and utilization in the PHC system of Ethiopia.

## Introduction

Globally, reproductive and maternal health promotion is an international pledge to invest in ensuring access to quality health services [1]. As a result, maternal mortality has declined substantially since the inception of the Millennium Development Goals [2], and striving to reduce the maternal mortality ratio to less than 70 per 100, 000 live births by 2030 [3,4]. About 95% of maternal deaths out of 287,000 women died due to pregnancy and childbirth related conditions occurred in Low- and Middle-Income Countries (LMICs) [2,4], which is empirical evidence for the existing inequalities in access to maternal healthcare service in LMICs [5]. Lack of equitable access to quality maternal health service is still a tragic public health agenda in LMICs that resulted in severe maternal morbidities and mortality [6]. Gender-related barriers impacted women's access to and use of health services in LMICs.

Gender refers to the socially constructed roles, behaviors, activities, and attributes that a given society considers appropriate for men and women [7,8]. It is a social construct defined by power relations at the individual, family, community, institutional, and structural levels in relation to one's perceived and/or personally experienced identity as a woman, man, girl, and boy [7]. As a social construct, gender pertains to the social meanings prevalent in a given society about the economic, social, and political roles, as well as the responsibilities, rights, norms, and expectations, associated with those identities. Barriers related to geographical distance or remoteness and poor infrastructure (roads) linked to safety concern [5,6,9–12]; cultural beliefs [5,6]; financial constraints for transportation [5,6,9,10]; lack of women's access to and control over resource; low women's decision-making autonomy and no voice in the health system [11–15]; inequitable social norms and expectations [10–13,15]; women having high workload and lack of division of labor with their male partners [10,13] are affecting women's maternal health service use LMICs.

Though the government of Ethiopia adopted the primary health care approach meanwhile in the 1970s, with significant development in service delivery models since 2003, and deployed thousands of female Health Extension Workers (HEWs) to ensure access to and deliver essential health services to the underserved communities [16], women's and girl's healthcare services uptake is still limited. Primary health care units (composed of primary hospitals, health centers, and health posts) represent the most accessible part of the health service delivery system and are responsible for the provision of essential health services [16]. With a diverse culture and gender norms in Ethiopia, women and girls face challenges in having access to equitable and

gender-responsive healthcare services. Women and girls are disproportionately affected in various aspects of their lives and health conditions, gender norms, societal expectations, and structural barriers intersect, resulting in disparities in health outcomes between men and women [17]. This gender inequality is particularly evident in rural areas, where women juggle multiple reproductive, community, and labor-intensive productive roles with less visibility and recognition [18].

The persistence of gender disparities existed as a result of social construction and patriarchal society in Ethiopia, the primary health care system is also affected by the existing social norms, despite significant efforts to improve healthcare access and outcomes, which calls for a comprehensive analysis to explore the gender dynamics influencing health-seeking behaviors, access to services and utilization in both agrarian and pastoral contexts. Thus, the aim of this study was to explore gender barriers to access to maternal and child health services in the primary health care setting in Ethiopia.

## Methods

### Ethical considerations

The research protocol was reviewed and approved by the Ethiopian Society of Sociologists, Social Workers and Anthropologists (ESSSWA) ethical review board (Ref No. ESSSWA/L/AA/0452/2023). The research team developed informed consent and assent forms for key informants and focus group discussants. Informed consent was obtained from all respondents before conducting interviews and discussions. Participation in this study was voluntarily without any implication to the study participants in their community. All the information from participants is kept confidential and presented anonymously.

### Study setting

Ethiopia is a low-income country located in the Horn of Africa [19] and the health system of the country has three tiers, namely, primary, secondary, and tertiary. The population of Ethiopia has an estimated population of more than 120 million of which a high majority live in rural areas [20]. Gender assessment was conducted in diverse contexts, including agrarian, agro-pastoral, and pastoral contexts, and encompassed nine woredas from eight regional states: Afar, Gambella, Oromia, Sidama, Central Ethiopia, South Ethiopia, Somali, and South West Ethiopia Peoples'. These agrarian, agro-pastoral, and pastoral contexts have differences in infrastructure access, health service access, livelihood, culture, and religious beliefs. The study involved Primary Health Care (PHC) facilities and surrounding communities from August to September 2023.

### Theoretical underpinnings and design

Despite the commitment of government and development partners to address gender disparities in primary healthcare settings, the influence of gender intersectionality on women's access to and use of essential health services is an untapped research agenda in Ethiopia. Unequal, unfair, ineffective, and inefficient gender inequities in the health and health system have remained a major public health challenge [21]. Understanding the reason for the unequal and unfair conditions and context-oriented intervention approach to curb the disparity is a critical agenda [21,22] that needs systematic exploration and program re-designing. Hence, we used a descriptive qualitative study design [23] to explore gender disparities in women's access to essential health services in the primary health care system in the agrarian and pastoral contexts of Ethiopia [21,22]. A multifaceted pragmatic meta-theoretical framework [24–26] was used to guide the study to understand gender disparities and their impact on access to and use of essential health services. The gender disparities were assessed mainly based on five USAID Transform Primary Health Care gender-analysis domains [27,28] that consolidated with other theoretical and implementation frameworks: namely cultural norms and beliefs; division of labor; access to and control over resources; patterns of power and decision; and health system. This study utilized the consolidated criteria for reporting qualitative studies framework [29] to ensure comprehensive reporting of the methods and results.

## Sampling methods

The maximum variation sampling technique was applied to recruit Key Informant Interviews (KIIs) and Focus Group Discussions (FGDs) participants to examine a diverse range of cases relevant to the gender analysis. Diverse experiences and opinions related to gender influence the transformation of gender- and social- norms and affect access to and use of Reproductive, Maternal, Newborn, Child, and Adolescent Health and Nutrition (RMNCAH-N) services by women, girls, and persons with disability in rural agrarian and pastoral contexts. The study participants were recruited through consulting kebele leaders, HEWs, and traditional birth attendants to identify appropriate persons for gender analysis. Community members, gender focal officers from nine woredas, the Ministry of Health, Ministry of Women and Social Affairs, who have active key roles and responsibilities RMNCHAH-N were eligible for the KIIs. Of these, targeted participants who were unable to give consent for participation due to physical and/or mental illness during the study period (August to September 2023) were excused from the study. KIIs were conducted with a wide array of different focal persons, including at the federal level (MoH and MoWSA) and a focal person from the regional health bureau (RHB), woreda health office, woreda gender focal person, service providers from health centers and health posts, HEWs, religious or clan leaders, Women's Development Union (WDU), adolescent girls out of school, adolescent boys in school, and married adolescent girls and boys. FGDs were conducted with six to twelve discussants, for women and men groups at kebele levels.

## Data collection methods

Data collection guides were developed for KIIs and FGDs through reviewing relevant literature. The interviews and discussion guides were developed guided by the desk review of relevant documents including published articles and validation by relevant stakeholders. Semi-structured interview guides were used to collect qualitative data through KIIs with experts and FGDs with representatives from government organizations, non-government organizations, community, and service providers. Experienced and trained qualitative research assistants were deployed to collect qualitative data and take handwritten notes. Relevant data were extracted from relevant policy and program documents using a data extraction template and complemented by the primary study. All the interviews were conducted using local dialects and took an estimated 45–60 minutes for the interview and up to 90 minutes for focus group discussions at their vicinity and convenient time. HEWs, TBAs and kebele leaders helped the data collectors to identify the entry points and access the FGD discussants.

## Data management

Interviews and focus group discussions were audio recorded. The audio records were listened to and transcribed according to the verbatim of the study participants. Transcripts were translated from the local dialect to English following the recommended approaches from existing literature [30,31]. Then transcripts were coded using ATLAS.ti software. The data analysis was conducted using both inductive and deductive techniques. The coded data underwent a blend of content [32] and thematic analysis [33,34] to identify recurring themes and patterns regarding gender dynamics affecting access to RMNCAH-N services. The data analysis was guided by USAID's five gender domains [28] such as culture, norm, and belief; division of labor, pattern of power on decisionmaking, access to and control over resources, and health system-related barriers.

## Results

A total of 87 key informant interviews (male = 49 and female = 38), representing health institutions at different levels, were included. This comprised focal persons from the Ministry of Health and Regional Health Bureaus (n = 7), Woreda Health Offices (n = 9), and Woreda gender focal persons (n = 9). Eighteen FGDs (n = 134) were conducted with six to twelve discussants from different groups, including women and men at kebele levels (Table 1).

**Table 1. The summary of study participants in KII and FGDs from August to September 2023.**

| Region | Woreda | Setting | | | FGD | | KII |
|---|---|---|---|---|---|---|---|
| | | Pastoral/Agro-pastoral | Agrarian | Total | Male | Female | |
| Federal level | - | | | | – | | 1 |
| Oromia | Dire | ✓ | | 1 | 10 | 9 | 11 |
| | Seka | | ✓ | 1 | 8 | 7 | 8 |
| Central Ethiopia | Weredijo | | ✓ | 1 | 8 | 5 | 9 |
| South Ethiopia | Dasenech | ✓ | | 1 | 7 | 7 | 10 |
| Afar | Chifra | ✓ | | 1 | 9 | 8 | 11 |
| Southwest | Benchi | | ✓ | 1 | 8 | 8 | 8 |
| Somali | Deghabur | ✓ | | 1 | 6 | 6 | 11 |
| Sidama | Bensa | | ✓ | 1 | 8 | 6 | 9 |
| Gambella | Lare | ✓ | | 1 | 7 | 7 | 9 |
| Total | 9 | 5 | 4 | 9 | 134 | | 87 |

The gender-related barriers and disparity in the primary health care setting are presented using the USAID's five gender domains framework (Table 2).

## Cultural norms and beliefs

Culture, norms and beliefs play a paramount role in influencing women's access to essential primary healthcare services in Ethiopia. The study participants emphasized that gender-related social norms, beliefs, and cultural practices in their communities often favor men than women. Women generally have the freedom to visit health facilities for Reproductive Maternal, Newborn, Child, Adolescent, and Youth Health and Nutrition (RMNCAYH-N), except for family planning. Unmarried adolescent pregnant women; women in polygamous marriages; and women living in rural areas have faced challenges in access to health services due to the presence of rigid gender norms. Cultural norms and religious beliefs restrict communities from having access to family planning and use in most settings of Ethiopia, mainly in pastoral contexts. Study participants, expressed the strongest reservations regarding family planning use, primarily citing religious reasons.

> *"...Birth control is condemned according to Afar culture and not allowed in Islam." - FGD, man, Afar*

> *"…Using family planning according to our religion is forbidden." -KII, religious leader, Jimma*

Moreover, these communities perceive that the utilization of family planning is against the basic essence of marriage childbearing and leads to promiscuity.

> *"…They believe that a girl marries a husband to have a child, so she should not use birth control."-KII, care provider, Afar*

Cultural beliefs played a more significant role in the lower ANC uptake. For instance, respondents from rural Borena mentioned that women tend to avoid seeking ANC because they believe pregnancy should be kept secret. They prefer not to reveal their pregnancy until their belly is visibly prominent, so this delays the initiation of ANC services.

> *"…if a woman gets pregnant, it is kept secret until six or seven months, known as 'Ulfi hin lalabamuu' [pregnancy should not be broadcast]. So, this practice can affect the ANC utilization or result in late initiation of ANC." - FGD, adult women, Borena*

**Table 2. Gender related barriers to primary health care service uptake in Ethiopia.**

| Domains | Barriers |
|---|---|
| Cultural norms and beliefs | • Religious prohibition of contraception, more pronounced in pastoral communities• Fear of promiscuity and infidelity<br>• Tribal demographic strategy to addressing conflict and resource scarcity<br>• Cultural secrecy and stigma around disclosing pregnancy early and labor onset |
| Division of labor | • Unequal division of labor<br>• Reproductive responsibilities overlooked<br>• Women's and girl's workload<br>• Domestic duties consume women's time (time poverty) |
| Control over resources | • Unequal distribution of assets (men predominantly own and control assets and resources; women's restricted financial autonomy)<br>• Intersection of gender norms and health-seeking behavior and access to essential information on RMNCAYH-N services |
| Patterns of power and decision-making | • Gendered decision-making dynamics, male dominancy<br>• Limited self-autonomy and conditional autonomy (restricted decision-making)<br>• Context-specific variations in power dynamics |
| Health system | • Lack of training on gender-responsive care and judgemental attitudes of health care providers<br>• Limited access to gender-responsive health information<br>• Lack of context-sensitive healthcare services |

These are deeply rooted in societal structures; existing hierarchal gender norms, cultural beliefs, and practices negatively influence women's access to and use of health services. According to a religious leader from the community:

*"…Our religion dictates that women should be under the authority of their husbands or families. Their roles are limited to domestic activities, and they are not permitted to engage in community activities without the permission of their husband or family."-KII, religious leader, Afar*

The prevailing unsupportive attitude towards family planning is predominantly influenced by demographic considerations and due to recurring conflict between tribes over resources. The cultural norm emphasizing procreation is so deeply ingrained that a husband may even choose to divorce his wife if he discovers that she is using contraceptives.

*"The primary reason behind many women's nonuse of modern contraceptives is the opposition from their husbands, who often do not permit them to use such methods. In some cases, women may resort to using contraceptives secretly without informing their husbands. However, if a husband becomes aware of his wife's contraceptive use, he may develop suspicions of infidelity and fear that she is engaging in extramarital relationships." (KII, WDA, Gambella)*

Unmarried adolescents' use of contraception is not socially acceptable.

*"… there is a cultural influence that if an unmarried girl uses birth control, she will become a prostitute." -KII, care provider, Afar*

Similarly, cultural norms and beliefs hindered unmarried adolescent girls' SRH service uptake. The community has a strong objection to unmarried girls seeking SRH services.

*".. if a girl went to the health center and asked for SRH services, the health providers are willing to provide the service. However, the parents are against premarital sex and do not allow their children to go to health centers for SRH services." -KII, adolescent girls, Gambella*

In addition, women did not want to disclose the onset of labor and believed other people's knowledge of a woman's onset of labor could prolong the labor. So that they want to keep it secret for the mother to give birth without trouble.

*"…Culturally, the onset of labor should be kept secret. If the labor is known to others, the women will experience prolonged labor." -KII, adolescent girl, Borena*

### Division of labor

There are distinct gender roles and responsibilities observed among women, girls, men and boys, and girls. These roles encompass domestic activities, caregiving, livestock-related tasks, child-rearing, marketing, and engagement in off-farm and nonfarm livelihood activities. Women carry a significant burden of household work. It appeared as a primary barrier that women's healthcare seeking has influenced due to their heavy workloads. Many women struggle to find the time to visit health facilities due to their extensive responsibilities at home.

*"…majority of household works are carried out by women and girls, leaving them with limited availability to visit health care facilities. Their visits to health facilities are often scheduled around the rare moments when they have some free time due to their demanding responsibilities." - KII, WDA, Afar*

Despite slow progress in sharing workloads with men and boys, women and girls continue to bear additional time and work burdens related to domestic and reproductive duties.

*"…Typically, women shoulder the majority of household chores and responsibilities. Men have relatively fewer domestic duties, with women being responsible for tasks such as cooking, washing clothes, childcare, water collection, wood gathering, and the care and supervision of cattle. Men and Boys' roles are mainly focused on tasks outside the house, such as tending to cattle and camels." -FGD, women, Dasenech woreda*

The extensive workload that women bear in their households hinders their ability and schedule to visit health facilities. "…*she forgets the appointed date due to workload at home.*" - FGD, adult women, Sidama
    Although women and girls are mainly engaged in routine and tedious domestic tasks, husbands play a crucial role in providing support to their wives, particularly during the delivery process. They ensure that their wives are taken to the health center, offer necessary assistance such as blood donation when required, and accompany them for immunization and post-delivery care.

### Access to and control over resources

According to informants, men typically hold a more privileged position when it comes to accessing and benefiting from assets and services. Although women do have some access to assets, they often have limited control over how these resources are used. Women farmers have even faced challenges in controlling resources despite having access to them. Lack of access to resources and control over cash among women was negatively impacting women from use of RMNCAH services, and limited women's ability to seek health care services, purchase medications or cover other expenses.

*"...men typically have access to and control over resources, including land and livestock. Men are often regarded as the owners and primary decision-makers when it comes to these resources. Women are often the ones who access credit services, with their husbands serving as guarantors for the loans from the village savings and credit association. However, even in cases where women access credit services, it is often the men who have control over that money, where men typically hold more decision-making power and control over financial resources." - FGD, adult men, Gambella*

Although access to and control over resources is equal for men and women by principle, men had disproportionality control over the resources. One of the study participants reported that;

*"…It's obvious there is male dominance in any household; there are many households with both males and females having equal access to and control over their resources and assets. There are a lot of improvements in gender equality compared to previous times." -KII, adolescent boy, Halaba woreda*

### Patterns of power and decision-making

Decision-making varies substantially across contexts, with notable differences between agrarian and pastoral areas. Women typically seek health care services only when directed to do so by their husbands or when they or their children are severely ill. The majority of the participants in our community share this view.

*"…women's health is the ones women can make decisions… especially in matters like family planning when a common decision cannot be reached."-FGD, women, Seka Chekorsa woreda,*

Although women have made decisions for antenatal care, choice of childbirth place, and accessing PNC, both male and female participants agreed that women typically carried the responsibility and have limited decision-making power on resources. The key informant remarked that:

*"…after marriage, I hold the decision-making power in all aspects of our life. If my wife doesn't comply with my decisions, I may consider divorce. This pattern has been passed down through generations, with my father following the same tradition. Any potential change in these dynamics would require community consensus and increased awareness among its members."- KII, adolescent boy, Deghabur.*

In many cases, the husband or father is the one who makes the final decision regarding health care.

*"…if a woman wishes to visit a health facility, she is allowed to do so, but she must inform her husband before. As per Afar culture, respecting the husband's honor, it is customary for the wife to communicate her intention to visit the health center. Generally, women can receive the necessary care by discussing their needs with their husbands." - KII, married adolescent girl, Chifra*

Women and girls often require permission from their husbands or fathers to seek healthcare. However, there is a disagreement, especially in matters related to family planning. Both male and female respondents from the agrarian areas also emphasized that men play a crucial role in providing various forms of support, including financial assistance and accompanying women for RMNCAH-N services.

*"Women take the lead in making health care decisions, but they often consult with their husbands to consider all available options. Women primarily make health care decisions. Nevertheless, in the case of family planning decisions,*

*there should be mutual support and agreement between husbands and wives." - FGD, women, Buyo Kechema kebele, Seka woreda*

### Health system-related barriers

**Gender awareness.** Healthcare providers had gaps in gender awareness that might be either insufficiency or a total lack of in-service gender integration training interventions. This knowledge gap contributes to disparities in accessing and delivering quality health services through addressing the unique needs of women and the barriers faced. Participants emphasized the need for onsite training as a primary solution, particularly in pastoral health centers and posts. A member of the WDA from the Afar region elaborated on this situation.

*"…Afar men desire to have more children, and as a result, they are generally opposed to their wives using birth control…nevertheless, some women choose to use birth control secretly. Since men are unsupportive of this practice, engaging in discussions with their husbands or obtaining their consent proves to be challenging. As a result, women often access these services independently and discreetly." - KII, WDA, Afar*

**Lack of access to gender-sensitive health information.** The study found that various sources provide information in the study areas, with HEWs, VHLs, health centers, radio, training by NGOs, community meetings, and churches being the most frequently cited sources among women, men, and adolescents. There are also disparities in access to accurate information, with men and youths mentioning social media as a source, while adolescents in rural areas and those with lower levels of education may face greater information gaps, leading to poor access and utilization of health care services.

*"Most of the time, we get information from VHLs. They go door to door, arrange mothers' forums, make campaigns in collaboration with the HEW and other NGOs (JSI and Sidama Development Association) about the HPV vaccine." - KII, service provider, Bensa woreda*

This can be attributed to the work of TBAs and VHLs who visit households and provide education about these services. During these visits, men are often away from home, so women can receive information and guidance about RMNCAH. In addition, the reluctance of many men in pastoral settings to accompany their wives to health centers and health posts, where information about RMNCAH-N is typically provided, further contributes to this gender gap in access to information.

*"…Men only get the chance if they are called for training. Relatively, women go for checkups, delivery, and vaccination, and this would give them the chance to have more access to health information than the man. Most men are either don't voluntarily visit health posts and centers or are in remote areas looking after their livestock." -KII, TBA, Degahbur woreda*

Generally, the lack of culture-friendly awareness creation for family planning in the communities caused a low uptake of family planning services by women.

**Judgmental attitude of healthcare providers.** Unmarried adolescents encounter negative attitudes from healthcare providers. Participants highlighted the difficulties young adolescents face when it comes to access to SRH services and use. According to key informants, adolescent girls in this community often do not support or approve of the use of SRH services. The judgmental attitude exhibited by certain healthcare providers serves as an additional deterrent for adolescents seeking SRH services.

*"…Our community is not happy to see adolescent girls use SRH services. Some care providers also judge you when you ask for the service."* -KII, adolescent girl, Bensa

**Lack of context-sensitive healthcare service.** Disparities persist in accessing friendly services that address the needs of women, adolescents, and persons with disabilities. The lack of gender-responsive and inclusive services exacerbates existing healthcare inequalities. Cultural and religious factors also contribute to their reluctance to be examined and assisted by male healthcare providers, as this goes against their cultural norms and religious beliefs. As a result, women seek health care only during severe complications that exceed the capabilities of TBAs. A care provider provided further insights into this situation.

*"...women avoid facility-based childbirth because of a pelvic examination; they do not want to disclose their private body. Even if they gave birth at a health facility, they want to leave soon before anyone sees them to avoid condemnation by the community."* - KII, care provider, Afar

As a result, women prefer TBAs over healthcare providers is a norm in pastoral communities. Then women mostly visited health facilities only if the delivery was complicated or if there was bleeding after birth.

*"…even if women acknowledge the importance of institutional delivery for safer and more medically supported childbirth, they view TBAs as a more comfortable and trusted option."* - FGD, women, Dasenech

## Discussion

Gender-related barriers to maternal health services in Ethiopia reflect a complex interplay of cultural norms, resource control, decision-making dynamics, and health system challenges, compounded by intersecting disadvantages—including gender, education, geography, and poverty-embedded across household, community, and institutional levels. These barriers are especially pronounced in pastoralist communities, where mobile lifestyles and limited service availability further constrain access. Cultural beliefs, such as religious prohibitions against contraception, stigma around pregnancy disclosure, and a preference for traditional practices, especially in pastoral communities, limit women's access to modern health services. Women's time poverty, arising from unequal labor division and domestic duties, combined with restricted financial autonomy due to male control of resources, further constrains their ability to seek care. Decision-making remains male-dominated, and women experience conditional autonomy that varies across agrarian and pastoral contexts. Within the primary health care systems, gaps in gender-responsive training, judgmental attitudes from providers, and insufficient context-sensitive services exacerbated these challenges.

This study highlights how women's limited autonomy—shaped by sociocultural norms and systemic health system gaps—significantly restricts maternal health access in rural Ethiopia, with pastoralist women facing even greater barriers due to mobility challenges and entrenched male gatekeeping. Similar patterns appear across low-income countries in sub-Saharan Africa, such as South Sudan and Mali, where male-dominated decision-making and cultural expectations limit women's healthcare use, compounded by economic constraints [35]. While Ethiopia has made progress, regional disparities remain, especially for pastoralist communities whose nomadic lifestyles and traditional gender roles pose unique challenges to accessing stationary health services [36]. These findings underscore the need for culturally sensitive, community-tailored interventions that address gendered power dynamics at household, community, and health system levels.

The existing cultural norms and beliefs impacted access to health care services and use. Norms, values, and religious beliefs around childbearing rooted in scriptural narratives adversely impacted health service uptake [15,37]. The culture and beliefs seriously affect unmarried women and girls' access to and use of health care [38]. The framing of gender as

a women's health issue, advanced through patriarchal structures, does little to elevate the status of women and promote a power balance that contributes to health inequity [39]. In addition, women do not want to go alone to health facilities causing low health service access in most areas of the country [40]. Religious and socio-cultural norms as well as gender stereotypes were important influences on the uptake and utilization of maternal health services, including facility-based delivery and contraception [41]. Low social support and perceived poor quality of health service influenced women's access to health care in which the women's agency has to negotiate their care experience [42].

In the present study, women's low autonomy in decision-making influences access to essential health services in the primary health care setting. Compared to previous studies, which emphasizes the dominant role of men and unilateral decision-making affects healthcare use by women in rural parts of Ethiopia [15,37,39], our findings add a contextual layer by illustrating how this autonomy is further curtailed by women's time poverty and financial dependency. Women need money for transportation and treatment and require permission from their husbands for medical care limiting access to health services [40]. In addition, overwhelmingly, women's workload in household duties and caring for family members negatively affect their access to and use of health services. This might worsen due to low literacy [38,40,43] and employment or engagement in paid jobs [43]. Men were predominant decision makers about women's health and healthcare seeking in both public and private spheres [39].

Although institutional support and motivation to healthcare providers improve the quality of health service delivery, women's gender preference and gender imbalance in the health sector leadership and women's decision-making affect healthcare and providers' responsiveness [39]. Lack of efforts to engage men in healthcare facilities, as well as the perception that healthcare facilities do not meet men's needs, highlight systems-level barriers to men's use of family planning services [37]. Despite not having holistic maternal health information which creates challenges in maternal health care utilization [15]. Health systems reinforce patients' traditional gender roles and neglect gender inequalities in health, women have less authority as health workers than men and are often devalued and abused; gender equality policies are associated with greater representation of female physicians, which in turn is associated with better health outcomes. Gender parity is insufficient to achieve gender equality. Institutional support and respect for nurses improve the quality of care, and women's empowerment collectives can increase healthcare access to and provider responsiveness [44].

The complexity and intersectionality of gender-related barriers identified in this study demonstrate that "one-size-fits-all" policy approaches are insufficient to address the diverse and context-specific challenges women face. While national gender equity policies provide a critical foundation, they must be complemented by targeted, culturally-sensitive, and localized interventions that tackle structural inequities and sociocultural norms. Community-level gender-transformative strategies—such as engaging men as partners, creating safe spaces for women, and delivering tailored health education for pastoralist populations—can help shift harmful norms. Strengthening female leadership in primary health care, integrating gender-sensitivity into provider training, and expanding mobile outreach and peer-support models can enhance service responsiveness and access in remote areas [45]. Ultimately, embedding gender equity principles into the design, implementation, and monitoring of health services is essential to address the unique needs of women, men, girls, and boys within the primary health care system. Without such integrated and practical efforts, gender-related barriers will remain deeply entrenched.

## Conclusions

Although many women are deployed and play a crucial role in providing health care in Ethiopia, gender-related barriers still negatively influence women's and girls' access to and use of health services adequately. These gender analysis findings guide the design of gender-intentional primary health care service delivery implementation strategies in both agrarian and pastoral contexts. Prioritizing gender-responsive approaches, and addressing gender disparities in the health system requires transformative changes in well-being, power dynamics, and structural aspects. Gendered socio-cultural factors significantly impact the utilization of essential primary healthcare services in Ethiopia. Gender-based power dynamics

contribute to poor RMNCAYH-N outcomes. Communities' culture, norms, values and religious beliefs, lack of gender awareness for healthcare workers, low level of women's and girls' agency, low representation of women in health sector leadership, and lack of culture-sensitive and women-friendly health services are some of the barriers related to gender.

Beyond gender-neutral policies, there is a pressing need for multi-level, gender-transformative strategies that intentionally integrate gender considerations into the design and implementation of all health programs. It is suggested to work with agrarian and pastoral communities to overcome socio-cultural and knowledge-based gender barriers that limit women's access to available health services while institutionalizing and tailoring national gender-integrated policies into primary healthcare services. The primary healthcare units should have a contextual gender-integration implementation framework and in-service capacity-building platform to address existing gender disparities and barriers to improve RMNCAH-N in Ethiopia. Therefore, approaches systemically addressing gender gaps in the provision of service and mitigating gendered barriers to service utilization improve the availability of and access to essential health services in Ethiopia.

## Acknowledgments

We would like to acknowledge Amref Health Africa and JSI for the overall administrative support, the study participants, data collectors, and supervisors for their willingness to give their time and information for this study.

## Author contributions

**Conceptualization:** Gizachew Tadele Tiruneh, Mesele Damte Argaw, Biruhtesfa Bekele, Metadel Tesfaye, Nebreed Fesseha, Chala Tesfaye, Sintayehu Abebe Woldie, Muluken Dessalegn Muluneh, Lidiya Tefera, Frank DelPizzo, Addis Tamire, Temesgen Ayehu, Dessalew Emaway, Misrak Makonnen.

**Data curation:** Agumasie Semahegn, Yohannes Mehretie Adinew, Samrawit Hagos Tiku, Nuru Hussien, Salsawit Shiferraw.

**Formal analysis:** Agumasie Semahegn, Yohannes Mehretie Adinew, Samrawit Hagos Tiku.

**Funding acquisition:** Sintayehu Abebe Woldie, Muluken Dessalegn Muluneh, Lidiya Tefera, Frank DelPizzo, Addis Tamire, Temesgen Ayehu, Dessalew Emaway, Misrak Makonnen.

**Investigation:** Samrawit Hagos Tiku.

**Methodology:** Agumasie Semahegn, Gizachew Tadele Tiruneh, Mesele Damte Argaw, Biruhtesfa Bekele, Metadel Tesfaye, Nebreed Fesseha, Samrawit Hagos Tiku, Addis Girma, Nuru Hussien.

**Project administration:** Addis Girma, Mebrie Belete, Chala Tesfaye, Mikiyas Teferi, Hillina Tadesse.

**Resources:** Addis Girma, Hillina Tadesse, Sintayehu Abebe Woldie, Muluken Dessalegn Muluneh, Lidiya Tefera, Frank DelPizzo, Addis Tamire, Temesgen Ayehu, Dessalew Emaway.

**Supervision:** Metadel Tesfaye, Nebreed Fesseha, Samrawit Hagos Tiku, Mebrie Belete, Chala Tesfaye, Mikiyas Teferi, Hillina Tadesse.

**Validation:** Agumasie Semahegn, Mesele Damte Argaw, Biruhtesfa Bekele, Nebreed Fesseha, Asmeret Gebrehiwet, Addis Girma, Mebrie Belete, Chala Tesfaye, Mikiyas Teferi, Hillina Tadesse, Nuru Hussien, Anne Austin, Salsawit Shiferraw, Sintayehu Abebe Woldie, Muluken Dessalegn Muluneh, Lidiya Tefera, Frank DelPizzo, Addis Tamire, Temesgen Ayehu, Dessalew Emaway, Misrak Makonnen.

**Visualization:** Agumasie Semahegn, Yohannes Mehretie Adinew, Mikiyas Teferi, Salsawit Shiferraw.

**Writing – original draft:** Agumasie Semahegn, Yohannes Mehretie Adinew, Mesele Damte Argaw.

**Writing – review & editing:** Agumasie Semahegn, Gizachew Tadele Tiruneh, Mesele Damte Argaw, Biruhtesfa Bekele, Metadel Tesfaye, Nebreed Fesseha, Samrawit Hagos Tiku, Asmeret Gebrehiwet, Addis Girma, Mebrie Belete, Chala

Tesfaye, Mikiyas Teferi, Hillina Tadesse, Nuru Hussien, Anne Austin, Salsawit Shiferraw, Sintayehu Abebe Woldie, Muluken Dessalegn Muluneh, Lidiya Tefera, Frank DelPizzo, Addis Tamire, Temesgen Ayehu, Dessalew Emaway, Misrak Makonnen.

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
