## [Editor Report · Decision Letter 0]

PGPH-D-24-02755

Gender disparities and barriers to access and use of essential health service in Ethiopia: designing primary health care through gender lens

Dear Dr. Semahegn

Thank you for submitting your manuscript to PLOS Global Public Health. After careful consideration, we feel that it has merit but does not fully meet PLOS Global Public Health’s publication criteria as it currently stands. Therefore, we invite you to submit a revised version of the manuscript that addresses the points raised during the review process.

The paper brings a description of the current challenges.  It includes some contextual factors and recognition of systemic and gendered barriers. 

There are elements that need addressing.

Given the paper is about a gender lens there is no mention of the gender framework or theories used to inform the thinking. Whilst there is mention that gender is a construct there is no critical underpinning. The subject matter is gender but there is no gender theory. Given the gender theories in the literature I feel this is a missing aspect of the paper.

The sample information is incomplete and I am not clear on the inclusion / exclusion criteria. 

The results section could be more clear. Whilst there are themes noted, it is not clear how many contributors contributed to each themes. The quotes show the themes, which may be enough. However,. there is nothing then constructed from themes.

The discussion section could be more clear and build on the citations. 

The grammar is not clear and in particular lines 14-26 need to be rewritten to give more clarity. It does not make enough sense.

The conclusion - ‘communities culture’ should be community’s culture or communities’ culture.

I suggest running this through a grammar check and addressing the conceptual gaps and clarity needed as noted

Please submit your revised manuscript by 09/03/2025 If you will need more time than this to complete your revisions, please reply to this message or contact the journal office at globalpubhealth@plos.org. Please include the following items when submitting your revised manuscript:

We look forward to receiving your revised manuscript.

Kind regards,

Dr Marion Lynch

Academic Editor

[

---

## [Decision Letter · Decision Letter 1]

PGPH-D-24-02755R1

Gender disparities and barriers to access and use of essential health service in Ethiopia: designing primary health care through gender lens

Dear Dr. Semahegn,

Thank you for submitting your manuscript to PLOS Global Public Health. After careful consideration, we feel that it has merit but does not fully meet PLOS Global Public Health’s publication criteria as it currently stands. Therefore, we invite you to submit a revised version of the manuscript that addresses the points raised during the review process.

The manuscript has been evaluated by one reviewer, and their comments are available below.

The reviewers have raised a few remaining of concerns. They feel that the discussion should contain a deeper reflection, given the sensitive nature of the topic. Could you please carefully revise the manuscript to address all comments raised?

We look forward to receiving your revised manuscript.

Kind regards,

Johanna Pruller, Ph.D.

PLOS Staff Editor

Journal Requirements:

Additional Editor Comments (if provided):

Reviewers' comments:

Reviewer's Responses to Questions

**Comments to the Author**

1. If the authors have adequately addressed your comments raised in a previous round of review and you feel that this manuscript is now acceptable for publication, you may indicate that here to bypass the “Comments to the Author” section, enter your conflict of interest statement in the “Confidential to Editor” section, and submit your "Accept" recommendation.

Reviewer #1: (No Response)

2. Does this manuscript meet PLOS Global Public Health’s publication criteria ? Is the manuscript technically sound, and do the data support the conclusions? The manuscript must describe methodologically and ethically rigorous research with conclusions that are appropriately drawn based on the data presented.

Reviewer #1: Yes

3. Has the statistical analysis been performed appropriately and rigorously?

Reviewer #1: N/A

4. Have the authors made all data underlying the findings in their manuscript fully available (please refer to the Data Availability Statement at the start of the manuscript PDF file)?

Reviewer #1: Yes

5. Is the manuscript presented in an intelligible fashion and written in standard English?

Reviewer #1: Yes

6. Review Comments to the Author

Reviewer #1: I congratulate the authors for carrying out the study in a careful and comprehensive manner.

The article is structured and written in clear language. It addresses an important topic to be discussed at a global level.

I only present recommendations regarding the structuring of the discussion, which had already been requested for corrections. I recommend that the reflections be deepened, given that the authors address a sensitive topic such as gender inequality in terms of access to health care. There is still a lack of clarity regarding what constitutes a result and what the scientific literature reports. It would be interesting for the authors to make comparisons, including mentioning the scenario in which they are making the comparisons. To conclude, I also recommend more practical actions as an attempt to change this scenario, not just generalizing with national policies.

7. PLOS authors have the option to publish the peer review history of their article (what does this mean? ). If published, this will include your full peer review and any attached files.

**Do you want your identity to be public for this peer review?** For information about this choice, including consent withdrawal, please see our Privacy Policy .

Reviewer #1: No

---

## [Decision Letter · Decision Letter 2]

Gender disparities and barriers to access and use of essential health service in Ethiopia: designing primary health care through gender lens

PGPH-D-24-02755R2

Dear Dr. Semahegn,

We are pleased to inform you that your manuscript 'Gender disparities and barriers to access and use of essential health service in Ethiopia: designing primary health care through gender lens' has been provisionally accepted for publication in PLOS Global Public Health.

Best regards,

Julia Robinson

Executive Editor

Reviewer Comments (if any, and for reference):

Reviewer's Responses to Questions

**Comments to the Author**

1. If the authors have adequately addressed your comments raised in a previous round of review and you feel that this manuscript is now acceptable for publication, you may indicate that here to bypass the “Comments to the Author” section, enter your conflict of interest statement in the “Confidential to Editor” section, and submit your "Accept" recommendation.

Reviewer #1: All comments have been addressed

2. Does this manuscript meet PLOS Global Public Health’s publication criteria ? Is the manuscript technically sound, and do the data support the conclusions? The manuscript must describe methodologically and ethically rigorous research with conclusions that are appropriately drawn based on the data presented.

Reviewer #1: Yes

3. Has the statistical analysis been performed appropriately and rigorously?

Reviewer #1: N/A

4. Have the authors made all data underlying the findings in their manuscript fully available (please refer to the Data Availability Statement at the start of the manuscript PDF file)?

Reviewer #1: (No Response)

5. Is the manuscript presented in an intelligible fashion and written in standard English?

Reviewer #1: Yes

6. Review Comments to the Author

Reviewer #1: Os autores acrescentaram alguns estudos para dar maior sustentação a discussão já realizada anteriormente, porém houve pouco aprofundamento ao longo da seção discussão como recomendado.

7. PLOS authors have the option to publish the peer review history of their article (what does this mean? ). If published, this will include your full peer review and any attached files.

**Do you want your identity to be public for this peer review?** For information about this choice, including consent withdrawal, please see our Privacy Policy .

Reviewer #1: No
